# Profile of Service Use and Barriers to Access to Care among Brazilian Children and Adolescents with Autism Spectrum Disorders

**DOI:** 10.3390/brainsci12101421

**Published:** 2022-10-21

**Authors:** Beatriz Araripe, Cecilia Montiel-Nava, Daniela Bordini, Graccielle R. Cunha, Gabriela Garrido, Sebastián Cukier, Ricardo Garcia, Analia Rosoli, Daniel Valdez, Sheila C. Caetano, Alexia Rattazzi, Cristiane S. Paula

**Affiliations:** 1Departamento de Psiquiatria, Universidade Federal de São Paulo (UNIFESP), São Paulo 04017-030, SP, Brazil; 2Department of Psychological Sciences, Universidad de Texas Rio Grande Valley, Edinburg, 78539 TX, USA; 3Universidad de la República, Montevideo 11200, Uruguay; 4PANAACEA, Programa Argentino para Niños, Adolescentes y Adultos con Condiciones del Espectro Autista, Buenos Aires B1640EMQ, Argentina; 5Universidad de Chile, Santiago 8330015, Chile; 6Organización Estados Iberoamericanos Para la Educacion, la Ciencia y la Cultura (OEI), Santo Domingo 10108, Dominican Republic; 7Facultad Latinoamericana de Ciencias Sociales (FLACSO), Buenos Aires C1026AAC, Argentina; 8Programa de Pós-Graduação em Distúrbios do Desenvolvimento, Centro Mackenzie de Pesquisa Sobre a Infância e Adolescência, Universidade Presbiteriana Mackenzie (UPM), São Paulo 01302-907, SP, Brazil

**Keywords:** autism spectrum disorder, low and middle-income countries, cross-cultural, healthcare utilization, treatment barriers, child

## Abstract

Delayed diagnosis and a lack of adequate care for people with autism spectrum disorder (ASD) are related to worse outcomes and quality of life. This study aimed to identify the profile of service use, barriers to access care, and factors related to those barriers in Brazilian families with children with ASD. A total of 927 families with children with ASD (3–17 years) from five Brazilian regions completed an online version of the Caregivers Needs Survey. Results showed that the most used services were behavioral interventions and pharmacotherapy, while the most used professionals were neurologists, nutritionists, speech therapists, psychiatrists, psychologists, and pediatricians. The main barriers included waiting lists, costs, and the absence of services or treatment. Service use varied according to age, the region of residence, type of health care system used, and the parents/caregivers’ education. Access to behavioral interventions was more frequent among users of the private system/health insurance and families whose caregivers had higher education. The absence of specialized services/treatments was less frequent among residents of state capitals and families whose caregivers had higher levels of education. This study highlights how families with children/adolescents with ASD in Brazil face significant barriers to access care related to sociodemographic factors.

## 1. Introduction

Autism Spectrum Disorder (ASD) comprises a group of early-onset neurodevelopmental deficits in childhood, and its clinical manifestations occur in the areas of social interaction, verbal/nonverbal communication, repetitive behaviors and sensory impairment [1], with symptoms and signs identifiable early in life [2].

Children with ASD face more unmet specialty and therapy care needs in comparison with individuals with other developmental disabilities [3]. Lack of adequate assistance for people with ASD often has a negative impact on social, educational, and family spheres and, consequently, on society as a whole [4].

There are high direct/indirect financial costs related to ASD; Considering the lifespan of an individual with ASD with intellectual disability, in the United Kingdom the costs have been estimated to be approximately £1.5 million (US $2.2 million), and $2.4 million in the USA; while among those without intellectual disability, estimates are £0.92 million (US $1.4 million) in the UK and US $1.4 million in the USA [4]. Despite these high costs, non-treatment is known to lead to even higher costs, as well as more detrimental short and long-term consequences for the individual and society that warrant the provision of comprehensive assistance to this population. The provision of a greater number of professionals in the public health system trained in evidence-based treatments would help to remove barriers to access to care for individuals with ASD [5].

The mapping of the treatment and services received, as well the identification of the barriers to obtaining them, are important for understanding the current situation [6,7] and for planning public policies for the ASD population; however, in Brazil, there is almost a complete lack of data in this area.

The aims of this study were to: (1) describe the profile of the use of health and education services by children and adolescents with ASD in the five Brazilian regions; (2) describe the barriers to accessing these services and the financial impact of ASD on the family; and (3) identify family sociodemographic characteristics related to the main treatments received and the main access barriers.

## 2. Material and Methods

This study is part of a cross-sectional multisite study comprising samples from six countries of the Latin American Network for Autism-REAL: Argentina, Brazil, Chile, Uruguay, Venezuela and Dominican Republic. In this paper, data from the Brazilian sample was used [8,9].

### 2.1. Sample

Participants were parents/caregivers of people with ASD from the five Brazilian regions who voluntarily agreed to participate in an online survey. The inclusion criteria were parents or caregivers aged over 17 who were responsible for a child diagnosed with ASD.

At the end of the data collection, the sample comprised 1200 families. After excluding those with missing essential data, such as place of residence or the age of the person with ASD, there were a total of 1168 families. Families with children/adolescents under three years or over 18-year-old were then excluded, leaving a total of 927 families from all five regions of Brazil. Most of participants came from the Southeast region (53.2%). The average age of the children with ASD was 7.5 years, and most subjects were male (83.1%) (Table 1).

Children were younger in the Northeast of the country (*p* < 0.01). Most informants were mother/stepmother (83.4%) who had completed higher education or a postgraduate/specialization qualification (71.6%). This profile was homogeneous among all Brazilian regions. Caregivers reported that most children presented mild to moderate symptoms (88.2%); used complex sentences and phrases (32.1%); had mild intellectual developmental delay (33.7%) and nearly half had behavioral problems. One third of the sample had to travel at least 50 km to obtain a diagnosis of ASD, being particularly problematic in the north region.

### 2.2. Instrument

The structured questionnaire used in this survey, the Caregivers Needs Survey, was developed by Amy Daniels and Autism Speaks [10]. It contains multiple-choice questions that are intended to assist in the initial mapping of the situation in countries that do not yet have a well-established policy on ASD support. It comprises four sections: (1) sociodemographic characteristics, (2) characteristics of the person with ASD, (3) use of services and treatments, (4) perception of parents/caregivers about barriers to access, challenges, priorities, and impact on the family.

The English-version of the questionnaire was translated into Portuguese by a child and adolescent psychiatrist and revised by a psychologist, both bilingual ASD specialists. This version was sent to a third bilingual professional who performed the back-translation and cross-cultural adaptation of the questionnaire, and after a few adjustments, the first Brazilian version entitled *Questionário para investigação de necessidades das pessoas com autismo segundo seu cuidador*, was produced. This version was then sent to two mothers of children with ASD to identify any words/sentences that were difficult to understand or possible inappropriate or stigmatizing terms. Some minor suggestions, all concerning the translation, were sent to the two individuals responsible for the back-translation/cross-cultural adaptation of the questionnaire who incorporated them into the final version.

### 2.3. Ethical Procedures and Considerations

All subjects gave their informed consent for inclusion before they participated in the study. The study was conducted in accordance with the Declaration of Helsinki, and the protocol was approved by the Ethics Committee of the Federal University of São Paulo (Project identification code: CAEE 48418515.7.00005505).

The study was advertised using various social networks managed by professionals in the area and family members. We also made direct contact with mothers and professionals who managed blogs/family mailing lists. General information about the study was provided as part of the call for volunteers, and access to the link to the questionnaire was open for three months.

Those interested in participating entered the MySQL program homepage. At the initial entry, they were presented with the informed consent terms and only after agreeing with these in the given virtual space, they received access to the questionnaire. Therefore, all participants provided written consent.

### 2.4. Statistical Analysis

Data were collected via MySQL software and the final dataset was converted for use with SPSS Version 17.0 (SPSS Inc., 2008, Chicago, IL, USA) for descriptive and inferential statistical analyses. Categorical variables were analyzed using the Chi-square test.

For the identification of factors related to the main services received and the main barriers to access to treatment, logistic regression models were used in the following manner: (i) variables with *p*-values < 0.20 in the bivariate analysis were selected for inclusion in the model, (ii) the variables that entered the model in the previous step and statistically significant (*p* ≤ 0.05) or borderline (0.10 > *p* > 0.05) were retained in the multivariate model, (iii) subsequently, variables that did not enter the model in the first stage (*p*-values ≥ 0.20 in the bivariate analysis) were introduced in the multivariate model and retained if their *p*-values were significant or borderline [11]. Considering the four family sociodemographic variables tested in the multivariate model (age group of children/adolescents with ASD; location [state capital or countryside]; type of health system used by the family; and education of parents/caregivers/informants), we assumed that there could be collinearity between the health system and the education of the parents/caregivers/informants. Thus, before building each of the multivariate models, a Cramer’s phi test was performed where a result > 0.60 would indicate the presence of collinearity, and one of the variables would have to be removed from the initial model.

## 3. Results

### 3.1. Use of Health Services

In Table 2, we can see that in the whole country, and similarly in the five Brazilian regions, approximately 1/5 of the individuals with ASD had never received treatment for ASD, with 20.7% of the whole sample receiving no treatment at the time they completed the survey, and individuals from the three less developed regions (North, Northeast, and Midwest) were even less well served.

Among those 20.7% in treatment, behavioral intervention was the most used (32.4%) health service, followed by pharmacotherapy (28.1%). In respect of lifetime use of services, a similar result was observed. However, the use of behavioral interventions (*p* = 0.15) was similar across all regions, while pharmacotherapeutic/drug treatment was less frequent in the northern region compared to the general average and in relation to the other regions of the country (*p* = 0.02) (Table 2).

### 3.2. Health Professionals

Neurologists/neuropediatricians (56.7%) were the most consulted health professional for the children and adolescents in the sample in the previous 12 months, followed by nutritionists (46.4%) and speech therapists (41.8%). The analysis of lifetime use of health professional showed that speech therapists are the most used in all regions of Brazil (69.1%), followed by neurologists/neuropediatricians (64.6%) and psychologists (63.1%), with identified regional differences (*p* = 0.04). It is important to mention that data regarding the care provided by psychologists in the previous 12 months was not collected (Table 2).

### 3.3. School Profile

Most of the children/adolescents (85.6%) in the sample attended mainstream school, with some regional variations, being highest in the northeast (93.7%) and lowest in the north of the country (72.2%). The majority of the study participants were attending private schools (54.7%), with some regional differences. The Northeast (71.7%) and Southeast (54.3) had a higher proportion of children attending private school while the North (41.7%), Midwest (43.6%) and South (44.0%) had a higher proportion of children attending public schools. Only a low percentage of children and adolescents were not attending any kind of school at the time of the study (4.2%), except for the Northeast region (16.7%). Additional educational support—to which they are entitled by law- was reported only by 61.1% of the participants, with the lowest report in the northern region (50.0%). All regions of the country reported low use of specialist teachers (7.7%), with 33.7% of lifetime use, and the northeast region reporting 23% (Table 2).

### 3.4. Insurance Type

Approximately one quarter of the children and adolescents with ASD in this sample exclusively used the public health system—SUS. In contrast, private health insurance was the most used health system (55.9%) with similar percentages in the five Brazilian regions (Table 3).

### 3.5. Barriers to Accessing Services

In general, the main barriers to access to treatment cited by the informants were lack of access (waiting lists) (59.6%) and the costs of treatment or services (38.7%). These barriers were similar across the country, except in respect of lack of access, which was highest in the northern region (reported by 85.2% of informants) and lowest in the southern region, with a frequency of 45.2%. Among parents/caregivers, 46.4% reported “always” or “often” feeling frustrated when seeking assistance for their child/adolescent with ASD, with 43.1% reporting “sometimes” feeling like this, with uniformity between the regions of the country (Table 3).

### 3.6. Economic Impact on Families

When considering the whole country, 48.7% reported suffering financial losses, 43.6% had a family member who had reduced their working hours, and 36.6% had a family member who had completely stopped working due to having a person with ASD in the family. These three factors related to the financial impact of ASD were similar across the five regions, except for a family member completely stopping working, which in the Midwest region, with 25.0%, was below the national average (Table 3).

### 3.7. Sociodemographic Factors, Use of Treatments, and Barriers to Accessing Services

The type of health system used by the family and the educational level of the participants were initially tested to avoid collinearity. A strong relationship between the variables was predicted, as lower income populations more often use the public health system. As expected, there was an association between the two variables (*p* < 0.01; 95% CI 3.14–7.10), but no collinearity (Cramer’s phi = 0.322), which allowed the entry of both variables in the initial multivariate models.

Table 4 presents the five multivariate models created to test the associations between familial sociodemographic factors and (1), the most commonly used treatments, and (2), the main barriers to obtaining them. As the two most frequently received types of treatment were equal in the current period and in the past (behavioral intervention/modification and pharmacotherapy), it was decided to present the associated factors only in the present.

Commonly used treatments: the first multivariate model indicated that those families who exclusively used the public health service (*p* < 0.01; 95% CI 0.36–0.75) and whose parents/caregivers had lower education (*p* < 0.01; 95% CI 0.38–0.78) had a reduced chance of accessing behavioral intervention/modification. regarding drug treatment, the final multivariate model revealed that younger children (three to six years old) received less medication than older children/adolescents (seven to 18 years old) (*p* < 0.01; 95% CI 0.43–0.77) (Table 4).

Barriers to accessing services: all four sociodemographic factors investigated in this study were associated with a waiting list as a barrier. Younger children (from 3 to 6 years old; *p* < 0.01; 95% CI 1.21–2.46); public health users (*p* < 0.01; 95% CI 1.11–2.74) and children of less-educated parents/caregivers (*p* = 0.04; 95% CI 1.40–3.56) were more exposed to this barrier, while those living outside the state capital (countryside) reported a smaller problem with waiting lists than residents of in the capitals (*p* = 0.04; 95% CI 0.47–0.99). The cost of treatment, another listed barrier to accessing services, showed no statistical significance with the only remaining factor in the final logistic regression model, health insurance/private user (*p* = 0.08; 95% CI 0.56–1.04). Finally, in respect of the third barrier of access to treatment—the lack of specialized services/treatment for children/adolescents with ASD—there was greater difficulty among families in the countryside (OR: 1.7; 95% CI: 1.26–2.32) and whose parents/caregivers had a lower level of education (OR 0.71; 95% CI (0.51–0.98). No differences were identified between families using exclusively the public health system and those using private services/insurance (Table 4).

This is the first study investigating the profile of service use and the barriers to care in Brazilian families with children/adolescents with ASD, as well as the financial impact of ASD on these families. Compared to the Brazilian average educational level, mothers/caregivers in this sample were highly educated: 38.1% had university degree and 33.5% had a Postgraduate/Specialization, in comparison with the 15.3% of Brazilians aged >25 in the population who have completed higher education. Additionally, most families had health insurance and most children/adolescents attended private schools. Data collection was through an online survey, meaning that caregivers without internet access were excluded. The sample comprises individuals from the middle and upper strata of the population and is not therefore representative of the general population. However, this is the first attempt at collecting this type of data, and despite these limitations in respect of the sample, the results help to highlight the lack of services in the country and represent valuable information with significant epidemiological implications.

Firstly, we highlight that currently, 21% of the Brazilian sample didn’t receive treatment, even knowing that individuals with ASD need support in different levels and domains [1]. On the other hand, behavioral interventions (current and lifetime) were the most used type of treatment. Behavior modification/intervention, like applied behavior analysis (ABA), are recommended as the first line of treatment worldwide [12,13,14,15,16]. Although behavioral interventions were the most common treatment, it is important to note that most of participants (almost 70%) did not receive any intervention of this type. The second most common treatment was pharmacotherapy. Research studies indicate that appropriate use of medications for individuals with ASD can help to reduce a wide range of serious, challenging behaviors and/or medical conditions that may interfere with daily life. The most common targets for pharmacologic intervention are comorbid conditions (e.g., mental health problems) and other features (e.g., aggression, and hyperactivity) [14,15,16,17].

Neurologists, nutritionists, speech therapists, psychologists and psychiatrists were the most frequently consulted professionals. Even though pediatricians are the second largest group of medical specialists in the country- accounting for 40,000 out of a total of 382,000 [18]—they are less consulted than neurologists and other health professionals in respect of ASD. Many national and international initiatives have taken place in recent decades to try to involve pediatricians more in identifying and treating children/adolescents with ASD. One of the most commonly reported barriers is in respect of the establishment of the ASD diagnosis. In an Australian study, 58% of pediatricians and child psychiatrists were found to have failed to diagnose ASD in a process to screen for children who qualify for specialized educational support and to obtain health care allowance [19]. However, there has been some progress in this area, such as that reported in a Brazilian study which found that training pediatricians to recognize ASD the early signs of the condition had a positive impact on diagnoses [20].

The Brazilian Inclusion Law has regulated the inclusion of children with ASD in mainstream schools since 2015, but even now, many children are still attending special schools, and this remains a controversial subject [21,22]. Despite being entitled by law to receive additional educational support, only 61.1% reported obtaining it. Moreover, less than 10% reported having a specialist educator. Across the country, a total of 4.6% of participants with ASD did not attend school. These findings are disturbing since the educational decisions that may be made due to lack of support not only affect the academic destiny of children, but also directly impact their personal destiny and their own development in a community that may marginalize difference and heterogeneity [23].

According to data from the Brazilian Institute of Geography and Statistics—IBGE, a minority (27.9%) of the Brazilian population has health insurance, with lower rates in the poorest regions of the country, notably in the northern (13.3%) and northeast (15.5%) regions. Overall, 71.1% of Brazilians use the public health system [24]. The data in our study is not in line with these rates as most of the sample had private insurance and used private health services (55.9%) rather than using the public health system (26.1%). This is a result of the general profile of the participants, who had an educational level that was much higher than the average of the Brazilian population. However, it shows that even for those with higher levels of education and more financial resources, finding services for their children with ASD is an ordeal, and offers some idea of how this journey might be for those with fewer resources. Our findings also showed that both public and private services for individuals with ASD are inadequately distributed across regions [25,26].

The biggest barriers found in this study were waiting lists, treatment costs, and the scarcity of specialized service, all of which are usually classified as structural barriers. The study also showed discrepancies between regions, mainly a lack of vacancies for care in the north and northeast regions. Recent surveys have shown that there are 650 institutions treating people with ASD in Brazil, the vast majority in the southeast, with 66.3% in the state of São Paulo. It is clear that the number of institutions is unevenly distributed among the five Brazilian regions, increasing the reported barriers to care in these areas [26]. These results depict a sad and unfair reality for the families of individuals with autism in Brazil, as in other low- and middle-income countries. Waiting for services means delaying the start of treatment, which in turn is related to worse outcomes. Whether this wait is based on simply not being able to afford services or the lack of available specialized services, the consequences are the same—delays in the start of treatment and a reduction in the child’s opportunities for better development.

Internationally, the most common barriers to treatment and/or service for people with ASD are a lack of specialized services and appropriate therapies, as well as, lack of access to this type of service due to costs and/or waiting lists which are classified as shortage of infrastructure [10,27]; in some situations with minority-language speakers, language barriers are also a problem [28]. In addition to these reasonably well-established barriers, other studies have sought to investigate the sociodemographic factors involved in this process. Overall, these studies indicate that families with lower purchasing power and less education often have more difficulty accessing specialized services [29,30,31] as did our study.

Confronting these barriers to accessing services for a child with ASD can lead to a feeling of frustration in caregivers, with almost half of the participants stating that they often/always felt like this, while only 10% reported never being frustrated when seeking services/treatments. The results of the survey reflect how difficult it is to access ASD services/treatment in Brazil, as also reported in studies conducted in other countries. A study in Serbia with a similar methodology to our study showed that one third of family members felt frustrated in relation to getting services/treatment for their children with ASD [32]. In addition to this structural barrier, other barriers have been reported which, if removed, would greatly contribute to more effective care. These include: (i) a lack of knowledge about ASD among parents, teachers and health professionals; (ii) a lack of information about existing services; (iii) a lack of reliability among the available public services, and (iv) stigma [6,33]. The current study found that 78.0% of respondents reported seeking information on ASD via the Internet and 50.1% from other parents of children with ASD. These results are like those obtained by Pejovic-Milovancevic et al. [32] in Serbia.

The presence of a person with ASD usually has an economic impact on the family [4,34]. The results of this study indicate the extent of the financial impact of ASD, resulting from the high costs of the different treatments required over the course of an individual’s life, as well as from the impact on parental employment profiles.

This study also identified several factors predicting access to the most common health services, as well as the main barriers to accessing treatment. Behavior modification/intervention was concentrated in the families of caregivers with higher education using private health services and having health insurance. These results reflect the social inequality in the country, with those who have more financial resources or education having better access to evidence-based treatment [35].

Pharmacotherapy was less frequent in younger children than in children over 7 years of age, which agrees with international findings. In a multicenter study with data from 17 health centers in United States and Canada, only 1% of children with ASD under three years old used any medication, while 10% of children between 3–5 years of age, 44% of 6–10-year-olds, and 64% of 12–17-year-olds used one or more medication [36]. In summary, the younger the child, the less frequent the use of medication.

Finally, in terms of the barriers, younger children who were users of the public health system, with parents/caregivers with a lower level of education and who lived in the state capitals reported more difficulty in accessing treatment due to waiting lists. The underfunding of public health services means that there is a lack of specialist services and long waiting lists, which makes it very difficult for families with limited financial resources to access appropriate treatment [25,26,31].

Surprisingly, families from state capitals, where health services are usually concentrated [18], reported more difficulty in accessing treatment due to waiting lists. Careful analysis of the data showed that in parallel, families outside the state capitals reported more difficulty in accessing specialist services. Thus, our main hypothesis is that in smaller cities, access to the basic network should be easier, with the elimination of the large distances that make seeking treatment time consuming and costly. Additionally, in smaller towns, health and education networks are often more connected, maximizing overall services as well as the referral system. In contrast, when it comes to the more specialist service, the situation is the opposite, with most concentrated in the capitals.

Private/health insurance users seem to suffer the greatest impact related to treatment costs, as specialist services are often very expensive [37]. The national health system (SUS) provides universal free access for the entire population; therefore, users of the public system incur no costs. Despite the numerous difficulties faced by individuals with ASD and their families in Brazil, several advances for the population with ASD have occurred in recent years, such as: (1) the recent launch of guidance protocols about ASD for families and professionals [38,39,40]; (2) advances in the care of children/adolescents with ASD being one of the most common diagnosis of Child and Adolescent Psychosocial Care Center—CAPSI (the main source of healthcare for children and adolescents with mental health problem) [6,25]; (3) Laws, public documents and regulations that guarantee the right to inclusive education for children with ASD, the most recent being the “Special Education Policy in Respect of Inclusive Education of the Ministry of Education” [41].

Our study has two main limitations which should be noted. First, the use of a non-random sample with data collection via the internet. This methodological strategy had the benefit of allowing us to include participants from all over the country in our sample, but it is likely that this produced some bias since this method excluded those without access to the internet or who were not literate—probably the most deprived families from rural areas and small towns. Besides, the informants in our sample had a higher level of education than the average of the Brazilian population. Therefore, our results may have underestimated the difficulties in respect of finding treatment for children with ASD, especially evidence-based ones. Second, the ASD diagnosis was based exclusively on the caregiver’s report without direct evaluation of the participants by a health professional or being based on a medical record.

Despite these limitations, the study produced some important results that have significant implications, highlighting the struggles of families with a child with ASD to access care, and providing evidence that service provision should be a higher priority in the public health agenda of the country. The lack of access to services translates into more impairment of the individual, more burden on the families, and, in the longer term, more economic and social costs that might have been reduced or avoided by earlier interventions. The study delineates in detail a profile of health disparities related to sociodemographic characteristics that needs to be considered when planning and allocating resources to the different regions of the country.

## Figures and Tables

**Table 1 brainsci-12-01421-t001:** Family sociodemographic characterization and clinical profile of children with ASD (N = 927).

Family Features/ Region of Country	Brazil	North	Northeast	Midwest	Southeast	South	Difference between Regions of the Country
CHILD PROFILE							
Residential Region [N (%)]	927 (100.0)	42 (4.5)	180 (19.4)	60 (6.5)	493 (53.2)	152 (16.4)	-
Age (years) mean (SD) *	7.51 (4.13)	7.60 (3.95)	6.56 (3.63)	7.60 (3.87)	7.80 (4.32)	7.55 (4.09)	*p* < 0.01 **
Gender [N (%)]							
Male	770 (83.1)	27 (64.3)	151 (83.9)	43 (71.7)	407 (82.6)	142 (93.4)	*p* < 0.01
Female	157 (16.9)	15 (35.7)	29 (16.1)	17 (28.3)	86 (17.4)	10 (6.6)	
Levels of development [N (%)]							
Mild symptoms	433 (47.0)	13 (31.0)	80 (44.7)	28 (46.7)	244 (49.9)	68 (44.7)	*p* = 0.02
Moderate symptoms	380 (41.2)	16 (38.1)	79 (44.1)	27 (45.0)	189 (38.7)	69 (45.4)	
Severe symptoms	70 (7.6)	8 (19.0)	9 (5.0)	4 (6.7)	38 (7.8)	11 (7.2)	
Not known	39 (4.2)	5 (11.9)	11 (6.1)	1 (1.7)	18 (3.7)	4 (2.6)	
Verbal level [N (%)]							
Does not speak	184 (19.9)	12 (28.6)	41 (22.8)	10 (16.7)	92 (18.7)	29 (19.2)	*p* = 0.75
Uses only single words	128 (13.8)	9 (21.4)	26 (14.4)	7 (11.7)	66 (13.4)	20 (13.2)	
Uses 2- or 3-word phrases	130 (14.1)	7 (16.7)	24 (13.3)	8 (13.3)	71 (14.4)	20 (13.2)	
Uses 4- or 5-word sentences	186 (20.1)	7 (16.7)	39 (21.7)	12 (20.0)	95 (19.3)	33 (21.9)	
Uses complex sentences	297 (32.1)	7 (16.7)	50 (27.8)	23 (38.3)	168 (34.1)	49 (32.5)	
Intelligence Levels [N (%)]							
Severe delay	92 (10.0)	11 (26.2)	16 (8.9)	7 (11.7)	46 (9.4)	12 (8.0)	*p* = 0.05
Mild delay	310 (33.7)	10 (23.8)	67 (37.4)	20 (33.3)	155 (31.7)	58 (38.7)	
Average	217 (23.6)	14 (33.3)	39 (21.8)	15 (25.0)	122 (24.9)	27 (18.0)	
Above average	169 (18.4)	3 (7.1)	28 (15.6)	12 (20.0)	94 (19.2)	32 (21.3)	
Not known	132 (14.3)	4 (9.5)	29 (16.2)	6 (10.0)	72 (14.7)	21 (14.0)	
Behavioral Problems [N (%)]							
Yes	423 (45.6)	19 (45.2)	71 (39.4)	29 (48.3)	238 (48.3)	66 (43.4)	*p* = 0.33
No	504 (54.4)	23 (54.8)	109 (60.6)	31 (51.7)	255 (51.7)	86 (56.6)	
Distance traveled for diagnosis							
<25 km	395 (43.6)	16 (38.1)	74 (42.5)	27 (45.8)	218 (45.1)	60 (40.5)	*p* = 0.32
Between 25 and 50 km	178 (19.6)	6 (14.3)	28 (16.1)	8 (13.6)	108 (22.4)	28 (18.9)	
Between 50 and 100 km	109 (12.0)	4 (9.5)	21 (12.1)	6 (10.2)	60 (12.4)	18 (12.2)	
>100 km	224 (24.8)	16 (38.1)	51 (29.3)	18 (30.5)	97 (20.1)	42 (28.4)	
INFORMANT PROFILE							
Relationship with the child ^#^							
Mother/Stepmother	762 (83.4)	34 (82.9)	147 (82.6)	50 (84.7)	409 (83.8)	122 (83.0)	*p* = 0.60
Father	68 (7.4)	3 (7.3)	15 (8.4)	4 (6.8)	34 (7.0)	12 (8.2)	
Grandparents	31 (3.4)	-	9 (5.1)	1 (1.7)	15 (3.1)	6 (4.1)	
Others	52 (5.7)	4 (9.7)	7 (3.9)	4 (6.8)	30 (6.1)	7 (4.8)	
Level of education							
Completed primary education	18 (1.9)	-	2 (1,2)	1 (1.7)	9 (1.8)	6 (4.0)	*p* = 0.78
High school	244 (26.3)	12 (28.6)	48 (26.7)	11 (18.3)	141 (2.6)	32 (21.2)	
University Degree	310 (33.5)	14 (33.3)	57 (31.7)	20 (33.3)	172 (34.5)	47 (31.1)	
Postgraduate/Specialization	353 (38.1)	16 (38.1)	73 (40.6)	28 (46.7)	170 (34.5)	66 (43.7)	
Not known	1 (0.1)	-	-	-	-	-	

Note: * Standard Deviation; ** Bonferroni test.

**Table 2 brainsci-12-01421-t002:** Description of use of health and education services by type of approach and professional, current and past (N = 927).

Use of Services/Region of Country	Brazil N (%)	North N (%)	Northeast N (%)	Midwest N (%)	Southeast N (%)	South N (%)	Difference between Regions of the Country
Current Health Service							
Behavioral	300 (32.4)	6 (15.4)	54 (30.3)	25 (42.4)	164 (34.2)	51 (34.7)	*p* = 0.07
Pharmacotherapy	258 (28.1)	9 (22.0)	38 (21.1)	17 (28.3)	145 (29.8)	49 (32.5)	*p* = 0.13
Sensory Integration Therapy	159 (17.3)	5 (11.9)	38 (21.1)	12 (20.3)	77 (15.8)	27 (18.0)	*p* = 0.50
Social Skills Training	154 (16.8)	6 (14.6)	23 (12.8)	8 (13.6)	91 (18.7)	26 (17.2)	*p* = 0.39
Psychoanalysis	103 (11.2)	1 (2.4)	19 (10.6)	5 (8.5)	65 (13.3)	13 (8.6)	*p* = 0.13
Biomedical Treatment *	5 (8.2)	2 (4.9)	13 (7.2)	3 (5.1)	40 (8.2)	17 (11.3)	*p* = 0.48
Relational approach **	4 (3.7)	2 (4.9)	1 (0.6)	1 (1.7)	22 (4.5)	8 (5.3)	*p* = 0.10
None	192 (20.7)	9 (21.4)	49 (27.2)	16 (26.7)	91 (18.5)	27 (17.8)	*p* = 0.05
Lifetime Health Service							
Behavioral	394 (42.5)	14 (35.0)	71 (40.6)	33 (57.9)	214 (44.4)	62 (43.1)	*p* = 0.15
Pharmacotherapy	346 (37.6)	8 (20.0)	55 (30.6)	24 (40.0)	198 (40.3)	61 (40.9)	*p* = 0.02
Social Skills Training	231 (25.1)	8 (19.5)	36 (20.1)	21 (35.6)	129 (26.3)	37 (24.7)	*p* = 0.27
Sensory Integration Therapy	226 (24.5)	9 (22.0)	53 (29.4)	17 (28.8)	113 (23.0)	34 (22.7)	*p* = 0.63
Psychoanalysis	151 (16.5)	2 (5.0)	23 (12.8)	13 (22.0)	91 (18.6)	22 (14.8)	*p* = 0.19
Biomedical Treatment *	112 (12.2)	2 (4.9)	21 (11.7)	8 (13.6)	59 (12.0)	22 (36.7)	*p* = 0.11
Relational approach **	61 (6.7)	5 (12.5)	9 (5.0)	1 (1.7)	33 (6.8)	13 (8.7)	*p* = 0.02
None	201 (21.8)	7 (17.1)	41 (22.8)	15 (25.4)	103 (20.9)	35 (23.0)	*p* = 0.084
Current Health Professionals							
Neurologists/Neuropediatrician	527 (56.7)	17 (40.5)	93 (52.0)	35 (58.3)	294 (59.9)	87 (57.6)	*p* = 0.08
Nutritionist	427 (46.4)	17 (41.5)	85 (47.2)	30 (50.0)	239 (49.0)	56 (37.1)	*p* = 0.12
Speech therapist	384 (41.8)	17 (41.5)	75 (41.9)	24 (40.0)	213 (43.6)	55 (36.7)	*p* = 0.66
Psychiatrist	282 (30.9)	8 (19.5)	62 (35.0)	16 (27.1)	154 (31.8)	42 (28.0)	*p* = 0.28
Behavioral therapist	256 (27.8)	7 (17.1)	31 (17.4)	20 (33.3)	142 (29.0)	56 (37.1)	*p* < 0.01
Psychologist	-	-	-	-	-	-	-
Occupational therapist	-	-	-	-	-	-	-
Paediatrician	-	-	-	-	-	-	-
Lifetime Health Professionals							
Speech therapist	631 (69.1)	24 (57.1)	110 (61.5)	46 (76.7)	354 (71.8)	97 (63.8)	*p* = 0.14
Neurologists/Neuropediatrician	598 (64.6)	28 (66.7)	106 (58.9)	45 (75.0)	324 (65.9)	95 (62.5)	*p* = 0.19
Psychologist	584 (63.1)	21 (51.2)	112 (62.2)	42 (70.0)	326 (66.1)	83 (55.0)	*p* = 0.04
Pediatrician	545 (59.1)	24 (58.5)	94 (52.5)	42 (70.0)	302 (61.5)	83 (55.0)	*p* = 0.08
Occupational therapist	415 (45.2)	13 (31.0)	87 (48.6)	29 (48.3)	229 (47.0)	57 (37.7)	*p* = 0.07
Psychiatrist	327 (35.5)	9 (22.0)	39 (21.8)	27 (45.0)	222 (45.3)	30 (19.9)	*p* < 0.01
Behavioral therapist	277 (30.0)	10 (23.8)	40 (22.3)	28 (46.7)	158 (32.2)	41 (27.2)	*p* < 0.01
Nutritionist	161 (17.7)	10 (25.6)	34 (19.1)	9 (15.0)	76 (15.7)	32 (21.2)	*p* = 0.31
Current/Educational Service							
School Type							
Mainstream	715 (85.6)	26 (72.2)	149 (93.7)	51 (92.7)	376 (83.4)	113 (84.3)	*p* < 0.01
Special	79 (9.5)	3 (8.3)	4 (2.5)	4 (7.3)	53 (11.8)	15 (11.2)	
Does not attend	38 (4.6)	7 (19.4)	6 (3.8)	-	20 (4.4)	5 (3.7)	
School Category							
Private school	458 (54.7)	15 (41.7)	114 (71.7)	24 (43.6)	246 (54.3)	59 (44.0)	*p* < 0.01
Public school	331 (39.5)	15 (41.7)	38 (23.9)	31 (56.4)	179 (39.5)	68 (50.7)	
Other type of school	13 (1.6)	-	2 (1.3)	-	10 (2.2)	1 (0.7)	
Does not attend school	35 (4.2)	6 (16.7)	5 (3.1)	-	18 (4.0)	6 (4.5)	
Additional Educational Support							
Yes	428 (61.1)	13 (50.0)	73 (56.6)	38 (79.2)	218 (57.1)	86 (74.1)	*p* < 0.01
No/Not known	273 (38.9)	13 (50.0)	56 (31.1)	10 (20.8)	164 (42.9)	30 (25.9)	
Special Needs Teacher							
Yes	70 (7.7)	4 (10.0)	20 (11.2)	3 (5.1)	31 (6.4)	12 (8.0)	*p* = 0.27
Not	841 (92.3)	36 (90.0)	158 (88.8)	56 (94.9)	453 (93.6)	138 (92.0)	
Lifetime/educational service							
Special Needs Teacher							
Yes	310 (33.7)	12 (30.0)	41 (23.0)	25 (41.1)	171 (34.9)	61 (40.1)	*p* < 0.01
No	610 (66.3)	28 (70.0)	137 (77.0)	35 (58.3)	319 (65.1)	91 (59.9)	

* Gluten free, casein free, probiotics, etc.; ** Floortime, Son-Rise, RDI, etc.

**Table 3 brainsci-12-01421-t003:** Health System, barriers to access to treatment and financial impact of ASD (N = 927).

Health System Type	Brazil	North	Northeast	Midwest	Southeast	South	Difference Test between Regions of the Country
Exclusively health insurance	514 (55.9)	20 (47.6)	95 (53.4)	39 (66.1)	274 (55.8)	86 (57.7)	*p* = 0.63
Exclusively public	240 (26.1)	15 (35.7)	49 (27.5)	10 (16.9)	132 (26.9)	34 (22.8)	
Public/health insurance or private	106 (11.5)	4 (9.5)	21 (11.8)	4 (6.8)	57 (11.6)	20 (13.4)	
Exclusively private	59 (6.4)	3 (7.1)	13 (7.3)	6 (10.2)	28 (5.7)	9 (6.0)	
Barriers/Region of the country							
There was a waiting list							
Yes	384 (59.6)	23 (85.2)	80 (66.7)	23 (53.5)	216 (59.8)	42 (45.2)	*p* < 0.01
Not	204 (31.7)	2 (7.4)	33 (27.5)	18 (41.9)	106 (29.4)	45 (48.4)	
Not known	56 (2.8)	2 (7.4)	7 (5.8)	2 (4.7)	39 (10.8)	6 (6.5)	
The price/cost of service/treatment							
Yes	357 (38.7)	17 (42.5)	68 (38.0)	30 (50.8)	188 (38.1)	54 (35.5)	*p* = 0.23
Not	552 (59.8)	23 (57.5)	110 (61.5)	29 (49.2)	293 (59.4)	97 (63.8)	
Not known	14 (1.5)	-	1 (0.6)	-	12 (2.4)	1 (0.7)	
There was no service or treatment							
Yes	282 (30.7)	14 (34.1)	56 (31.3)	20 (33.9)	148 (30.3)	44 (28.9)	*p* = 0.30
Not	612 (66.5)	26 (63.4)	121 (67.6)	39 (66.1)	320 (65.4)	106 (69.7)	
Not known	26 (2.8)	1 (2.4)	2 (1.1)	-	21 (4.3)	2 (1.3)	
Not entitled to these services							
Yes	197 (21.4)	10 (25.0)	36 (20.1)	13 (22.0)	114 (23.2)	24 (15.8)	*p* = 0.74
Not	689 (75.7)	29 (72.5)	139 (77.7)	45 (76.3)	362 (73.6)	123 (80.9)	
Not known	27 (2.9)	1 (2.5)	4 (2.2)	1 (1.7)	16 (3.3)	5 (3.3)	
Could not get the information							
Yes	192 (20.9)	8 (20.5)	40 (22.3)	16 (27.1)	108 (22.0)	20 (13.2)	*p* = 0.06
Not	714 (77.7)	31 (79.5)	139 (77.7)	41 (69.5)	372 (75.9)	131 (86.2)	
Not known	13 (1.4)	-	-	2 (3.4)	10 (2.0)	1 (0.7)	
Other reason							
Yes	86 (9.5)	5 (12.8)	17 (9.5)	7 (12.1)	44 (9.1)	13 (8.6)	*p* = 0.54
Not	795 (87.5)	34 (87.2)	155 (86.6)	48 (82.8)	421 (87.3)	137 (90.7)	
Not known	28 (3.1)	-	7 (3.9)	3 (5.2)	17 (3.5)	1 (0.7)	
Frustration in seeking treatment							
How often did you feel frustrated when seeking services/treatments?							
Never	68 (8.7)	-	12 (8.6)	4 (7.7)	36 (8.4)	16 (12.8)	*p* = 0.22
Sometimes	336 (43.1)	13 (39.4)	65 (46.8)	20 (38.5)	182 (42.3)	56 (44.8)	
Often	239 (30.7)	10 (30.3)	38 (27.3)	21 (40.4)	141 (32.8)	29 (23.2)	
Always	122 (15.7)	10 (30.3)	23 (16.5)	7 (13.5)	61 (14.2)	21 (16.8)	
Do not know	1 4 (1.8)	-	-	-	10 (2.3)	3 (2.4)	
Impact on the family							
Your child caused a financial problem for the family.							
Yes	451 (48.7)	21 (50.0)	80 (44.7)	33 (55.0)	247 (50.1)	70 (46.1)	*p* = 0.57
Not	466 (50.3)	20 (47.6)	98 (54.7)	26 (43.3)	243 (49.3)	79 (52.0)	
Not known	9 (1.0)	1 (2.4)	1 (0.6)	1 (1.7)	3 (0.6)	3 (2.0)	
A family member reduced working hours							
Yes	404(43.6)	20 (47.6)	71 (39.7)	26 (43.3)	214 (43.4)	73 (48.0)	*p* = 0.58
Not	516 (55.7)	21 (50.0)	107 (59.8)	33 (55.0)	276 (56.0)	79 (52.0)	
Not known	6 (0.6)	1 (2.4)	1 (0.6)	1 (1.7)	3 (0.6)	-	
A family member stopped working							
Yes	339 (36.6)	16(38.1)	57 (31.8)	15 (25.0)	198 (40.2)	53(34.9)	*p* = 0.01
Not	580 (62.6)	24 (57.1)	120 (67.0)	45 (75.0)	292 (59.2)	99 (65.1)	
Not known	6 (0.6)	2 (4.8)	2 (1.1)	-	3 (0.6)	-	

**Table 4 brainsci-12-01421-t004:** Sociodemographic factors related to the most commonly health services and main access barriers (N = 970).

	Initial Logistic Multivariable Regression			Final Multivariable Logistic Regression		
(Independent Variables)	OR *	(95% CI) **	*p*	Adjusted OR	(95% CI)	*p*
	**Type of Health Services at present**					
**Intervention/behavioral modification**						
Age range (<6 years as reference)	0.96	(0.71–1.27)	0.76	-	-	-
Location (countryside as reference)	0.81	(0.61–1.10)	1.66	-	-	-
Health system (exclusively public as reference)	0.52	(0.36–0.75)	<0.01	0.52	(0.36–0.75)	<0.01
Informant Education (< education as reference)	0.57	(0.40–0.81)	<0.01	0.55	(0.38–0.78)	<0.01
**Pharmacotherapy**						
Age range (<6 years as reference)	0.57	(0.42–0.76)	<0.01	0.58	(0.43–0.77)	<0.01
Location (urban as reference)	1.18	(0.87–1.60)	0.39	-	-	-
Health system (exclusively public as reference)	0.76	(0.53–1.09)	0.14	-	-	-
Informant Education (<education as reference)	1.04	(0.74–1.47)	0.81	-	-	-
	**Barriers to treatment**					
**Waiting list**						
Age range (<6 years as reference)	1.72	(1.21–2.46)	<0.01	1.72	(1.21–2.46)	<0.01
Location (countryside as reference)	0.68	(0.47–0.99)	0.04	0.68	(0.47–0.99)	0.04
Health system (exclusively public as reference)	1.75	(1.11–2.74)	<0.02	1.75	(1.11–2.74)	<0.02
Informant Education (< education as reference)	2.23	(1.40–3.56)	<0.01	2.23	(1.40–3.56)	<0.01
**Costs**						
Age range (<6 years as reference)	0.98	(0.75–1.29)	0.89	-	-	-
Location (countryside as reference)	0.86	(0.65–1.13)	0.27	-	-	-
Health system (Exclusively public as reference)	0.80	(0.58–1.11)	0.19	0.76	(0.56–1.04)	0.08
Informant Education (<education as reference)	0.90	0.65–1.24)	0.51	-	-	-
**Lack of specialist service**						
Age range (<6 years as reference)	0.88	(0.66–1.17)	0.38	-	-	-
Location (countryside as reference)	1.72	(1.27–2.33)	<0.01	1.71	(1.26–2.32)	<0.01
Health system (exclusively public as reference)	1.07	(0.76–1.50)	0.72	-	-	-
Informant Education (<education as reference)	0.69	(0.49–0.98)	0.04	0.71	(0.51–0.98)	0.04

* OR = odds ratio; ** 95% CI = 95% confidence interval.4. Discussion.

## Data Availability

The Research data is not available in a public repository but can be share under request.

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
