# Peer review of "Profile of Service Use and Barriers to Access to Care among Brazilian Children and Adolescents with Autism Spectrum Disorders"

_brainsci, 2022, doi:10.3390/brainsci12101421_

Round 1
Reviewer 1 Report
In principle this research article is interesting, even if very geographic specific one.
My main issue is regarding the choice of the journal, as it seems not appropriate for this type of article.
Minor issue:
Limitation section: I do not agree with authors, because:
1. what is "convenience sample"?
2. the higher education level is not a limitation, at least there are other "autistic" families with lower education levels that did not participate for this reason (in example for having difficulties in understanding the questions?).
3. it is quite the same of the point 2. Please claim it in exclusion criteria.
4. why this point? the ASD diagnosis should be performed only by caregivers.
Author Response
Thank you for the opportunity to revise our original research manuscript entitled ‘Profile of Service Use and Barriers to Access to care among Brazilian Children and Adolescents with Autism Spectrum Disorders (brainsci-1924343)’
Please see the responses below:
1.1. In principle this research article is interesting, even if very geographic specific one.
My main issue is regarding the choice of the journal, as it seems not appropriate for this type of article.
Response:
The manuscript was submitted to the Special Issue entitled ‘Autism Spectrum Disorders (ASD) in Low- and Middle-Income Countries (LMICs) and among Immigrants and Minority Groups in High-Income Countries (HICs)’ under the section ‘Developmental Neuroscience’. We believe, therefore, that this article is an appropriate submission to this edition of Brain Sciences.
1.2. Minor issue:
Limitation section: I do not agree with authors, because:
1.2.1. what is "convenience sample"?
1.2.2 the higher education level is not a limitation, at least there are other "autistic" families with lower education levels that did not participate for this reason (in example for having difficulties in understanding the questions?).
1.2.3. it is quite the same of the point 2. Please claim it in exclusion criteria.
Response:
Convenience sample doesn't include a random selection of participants but selects subjects who are easy to contact in accessible places. In response to the reviewer’s suggestion, we replaced ‘convenience sample’ with ‘non-random sample’ in the manuscript (line 147).
To better explain the limitation related to ‘the higher education level of the participants’ we joined and updated the limitations. Please see line 148-153 in the revised version of the manuscript.
We prefer to keep this information in the limitations section because it was not an exclusion criterion but relates to the problem of not having data from more deprived families with lower education levels.
2.4. why this point? the ASD diagnosis should be performed only by caregivers.
Response:
Thank you for your question. We rephrased the limitation related to the ASD diagnosis to clarify this important issue.
Please see line 153-155 in the revised version of the manuscript, now described as the second limitation.
Reviewer 2 Report
The subject of the article refers to an important issue of child psychiatry, which is access to care and specialist treatment for children and adolescents with autism spectrum disorder (ASD) and the related barriers in various regions of Brazil. Based on the online survey, the authors obtained data from 927 families of children and adolescents diagnosed with ASD. From a methodological and ethical point of view, the research was carried out correctly. This also applies to the analysis of the obtained results and the resulting conclusions. It is also important that authors are aware of the limitations of the material obtained.
Nevertheless, the reviewed text contains a number of incorrect wording that require correction. Here are some examples:
- a slight change should be made in the title of the article, instead of 'to access' there should be 'to access care' (page 1, line 2),
- the entire text should be corrected by a native speaker because it contains a number of grammatical and stylistic errors (e.g. on page 2, line 57 there is: 'describe the barriers to access to these services', and it should be "describe the barriers to accessing these services'; on page 2, line 67 there is: 'to participate in an online data' and should be 'to participate in an online survey'). These types of errors appear throughout the text, including the titles of some tables.
Author Response
Thank you for the opportunity to revise our original research manuscript entitled ‘Profile of Service Use and Barriers to Access to care among Brazilian Children and Adolescents with Autism Spectrum Disorders (brainsci-1924343)’
Please see the responses below.
The subject of the article refers to an important issue of child psychiatry, which is access to care and specialist treatment for children and adolescents with autism spectrum disorder (ASD) and the related barriers in various regions of Brazil. Based on the online survey, the authors obtained data from 927 families of children and adolescents diagnosed with ASD. From a methodological and ethical point of view, the research was carried out correctly. This also applies to the analysis of the obtained results and the resulting conclusions. It is also important that authors are aware of the limitations of the material obtained.
Response:
Thank you for your review and support. Strong data about ASD from LMIC families is a real need, therefore it is much appreciated to see your support for our study.
2.1. Nevertheless, the reviewed text contains a number of incorrect wording that require correction. Here are some examples:
2.1.1 a slight change should be made in the title of the article, instead of 'to access' there should be 'to access care' (page 1, line 2),
2.1.2. the entire text should be corrected by a native speaker because it contains a number of grammatical and stylistic errors (e.g. on page 2, line 57 there is: 'describe the barriers to access to these services', and it should be "describe the barriers to accessing these services'; on page 2, line 67 there is: 'to participate in an online data' and should be 'to participate in an online survey'). These types of errors appear throughout the text, including the titles of some tables.
Response:
Thank you for your careful reading of the manuscript and your comments.
A native speaker revised the entire manuscript, including the specific points highlighted above: title, page 2 (lines 60 and 70), and the tables.
All changes are now in red in the revised version of the manuscript.
Please see the Revised manuscript attached.

Round 2
Reviewer 1 Report
Authors well addressed all my comments.